# HPLC/DAD, Antibacterial and Antioxidant Activities of *Plectranthus* Species (Lamiaceae) Combined with the Chemometric Calculations

**DOI:** 10.3390/molecules26247665

**Published:** 2021-12-17

**Authors:** Fabíola F. G. Rodrigues, Aline A. Boligon, Irwin R. A. Menezes, Fábio F. Galvão-Rodrigues, Gerson J. T. Salazas, Carla F. A. Nonato, Nara T. T. M. Braga, Fabrina M. A. Correia, Germana F. R. Caldas, Henrique D. M. Coutinho, Abolghasem Siyadatpanah, Bonglee Kim, José G. M. Costa, Adriana R. C. Barros

**Affiliations:** 1Northeast Biotechnology Network, Postgraduate Program in Biotechnology, State University of Ceará, Fortaleza 60740-000, Brazil; fabiolafer@gmail.com (F.F.G.R.); irwinalencar@yahoo.com.br (I.R.A.M.); hdmcoutinho@gmail.com (H.D.M.C.); adrianarol@gmail.com (A.R.C.B.); 2Health Unit, University Center Dr. Leão Sampaio, Juazeiro do Norte 63040-000, Brazil; narattmb@gmail.com (N.T.T.M.B.); fabrina@leaosampaio.edu.br (F.M.A.C.); germanafreire@yahoo.com.br (G.F.R.C.); 3Department of Biological Chemistry, Regional University of Cariri, Crato 63105-000, Brazil; fabiogalvao01@hotmail.com (F.F.G.-R.); timotygertor@yahoo.com (G.J.T.S.); carlaalvesbio@hotmail.com (C.F.A.N.); 4Department of Industrial Pharmacy, Federal University of Santa Maria, Santa Maria 97105-900, Brazil; alineboligon@hotmail.com; 5Ferdows School of Paramedical and Health, Birjand University of Medical Sciences, Birjand 9717853577, Iran; 6Department of Patology, College of Korean Medicine, Kyung Hee University, Seoul 02447, Korea; 7Korean Medicine-Based Drug Repositioning Cancer Research Center, College of Korean Medicine, Kyung Hee University, Seoul 02447, Korea; 8Experimental Biology Nucleus, University of Fortaleza, Fortaleza 60811-905, Brazil

**Keywords:** *Plectranthus*, antibacterial activity, antioxidant activity, synergism, DPPH, PCA

## Abstract

The increase in antibiotic resistance and the emergence of new bacterial infections have intensified the research for natural products from plants with associated therapy. This study aimed to verify the antibacterial and antioxidant activity of crude extracts of the genus *Plectranthus* species, being the first report on the modulation of aminoglycosides antibiotic activity by *Plectranthus amboinicus* extracts. The chemical composition was obtained by chemical prospecting and High-Performance Liquid Chromatography with diode arrangement detector (HPLC/DAD). The antibacterial activities of the extracts alone or in association with aminoglycosides were analyzed using the microdilution test. The antioxidant activity was evaluated by 2,2-diphenyl-1-picrylhydrazyl (DPPH) free radical scavenging. The phytochemical prospection allowed the flavonoids, saponins, tannins and triterpenoids to be identified. Quercetin, rutin, gallic acid, chlorogenic acid, caffeic acid, catechin, kaempferol, glycosylated kaempferol, quercitrin, and isoquercitrin were identified and quantified. The principal component analysis (PCA) observed the influence of flavonoids and phenolic acids from *Plectranthus* species on studied activities. Phytochemical tests with the extracts indicated, especially, the presence of flavonoids, confirmed by quantitative analysis by HPLC. The results revealed antibacterial activities, and synergistic effects combined with aminoglycosides, as well as antioxidant potential, especially for *P. ornatus* species, with IC_50_ of 32.21 µg/mL. Multivariate analyzes show that the inclusion of data from the antioxidant and antibacterial activity suggests that the antioxidant effect of these species presents a significant contribution to the synergistic effect of phytoconstituents, especially based on the flavonoid contents. The results of this study suggest the antibacterial activity of *Plectranthus* extracts, as well as their potential in modifying the resistance of the analyzed aminoglycosides.

## 1. Introduction

The progress of biotechnology, associated with health concerns such as the emergence of increasingly resistant microorganisms to conventional antimicrobials, as well as degenerative diseases, such as Alzheimer’s and Parkinson’s, which are associated with free radical-induced oxidative stress, have raised interest in both the development of new antimicrobial compounds and natural antioxidants obtained from medicinal plants [1,2,3].

Among the plant species most widely used in the treatment of diseases and ethnobotanical applications, the genus *Plectranthus* (Lamiaceae) should be highlighted; it is widely studied in ethnopharmacological and chemical terms, as it has a representative popular use in the form of teas, infusions, and syrups, especially in the treatment of digestive, dermatological, and respiratory diseases [4,5].

The *Plectranthus* genus includes about 300 species of herbs and shrubs native to tropical and warm regions worldwide [6]. In Brazil, several species of this genus are referred to as “boldo” and “mint”, and they are known for being able to successfully grow in different environmental conditions, tolerating from moderate to high water and nutrient deficits [7,8]. In addition, the genus is recognized for its biological activities, such as antimicrobial, anticancer, antiparasitic, repellent, immunomodulating activity, among others. Its bioactivities are related to its content of phenolic compounds, proving to be rich in flavonoids, especially flavones, flavonols and flavonones, and phenolic acids, such as *trans*-rosmarinic acid [4,9,10,11,12].

This study is a chemical approach to ethanol extracts of *Plectranthus amboinicus*, *Plectranthus barbatus*, and *Plectranthus ornatus* leaves, in which a comparison of the chemical profile was made, emphasizing the quantification of phenolic acids and flavonoids by HPLC/DAD; also, it describes the results from microbiological tests to verify the antibacterial effect and modulate the bacterial resistance to aminoglycosides and antioxidant potential. Therefore, it must be emphasized that this is the first report performed with extracts of *P. amboinicus* species as modulator of this class of antibiotics against pathogenic bacteria.

Moreover, this work provides a significant contribution to the chemical and biological knowledge of the *Plectranthus* species, to the validation of its popular use, and the availability of alternative sources, as well as the valorization of local biodiversity.

## 2. Results and Discussion

### 2.1. Chemical Characterization

The phytochemical prospection revealed the presence of flavonoids (flavones, flavonones, xanthones, chalcones and aurones), triterpenoids, tannins and saponins in the three extracts, as shown in Table 1. *P. amboinicus* was the only species that showed anthocyanins and anthocyanidins. However, this one did not indicate the presence of chalcones like the other species, which are in agreement with the literature [13]; it has a rich composition of flavonoids such as apigenin, cirsimartin, luteolin, salvigenin, rutin, and quercetin, as well as phenolic acids and tannins [14].

Studies show that in ethanolic, acetonic and aqueous extracts of *P. amboinicus,* leaves do not have anthocyanidins by phytochemical prospecting [15,16]. The amount and type of anthocyanins in vegetables are also influenced by some determinants, such as growing conditions, temperature, ultraviolet (UV) light exposure, salinity, drought, injury, and harvest method [17]. For this reason, comparison of anthocyanin contents between different crops may show different results.

Hiba et al. [18] studying the influence of zinc concentration and drought on the production of secondary metabolites of *P. amboinicus* showed a significant increase in anthocyanin content in individuals submitted to ZnSO_4_ treatment in 21 days (36%) compared to their respective controls, while in those subjected to water stress, there was a reduction of 33%.

Phenolic acids and flavonoids in extracts from fresh leaves of *P. amboinicus*, *P. barbatus* and *P. ornatus* were determined by the HPLC/DAD system using authentic standards and a total of ten compounds were determined from the samples. According to the results, the *Plectranthus* species contained comparable amounts of phenolic acids and flavonoids (Table 2). The results prove that the extracts contained among other compounds, as shown in Figure 1, gallic acid (Rt = 13.87 min, peak 1), catechin (Rt = 19.94 min, peak 2), chlorogenic acid (Rt = 24.15 min, peak 3), caffeic acid (Rt = 27, 45 min, peak 4), rutin (Rt = 39.83 min, peak 5), quercitrin (Rt = 42.28 min, peak 6), isoquercitrin (Rt = 45.06 min, peak 7), quercetin (Rt = 48.35 min, peak 8), kaempferol (Rt = 57.62 min, peak 9) and glycosated kaempferol (Rt = 66.31 min, peak 10). Many of those compounds have been shown to possess both antimicrobial and antioxidant activities in previous studies, thus contributing to the observed activities in such plant extracts [19,20].

Phenolic compounds are extensively used in botanical chemosystematic studies. The chemotaxonomic values of phenolic biomolecules such as these has been recognized in the plant kingdom [21]. Apart from the chemotaxonomic significance, its biological activities stand out as well as the role of phenolic compounds as indicators of the presence of metals in the leaves of different botanical genera [22].

There are considerable differences in the concentrations of phenolic compounds compared to the three species of *Plectranthus*. In EELPa (1.37%), there is the highest amount of chlorogenic acid (4.06 ± 0.03 mg/g), while in EELPb (1.91%) and EELPo (3.75%) the amount of caffeic acid was the highest (5.01 ± 0.50 mg/g and 9.76 ± 0.05 mg/g, respectively). In the EELPo extract, the constituents are in higher quantity, showing high concentrations quercitrin (4.93 ± 0.03), chlorogenic acid (4.51 ± 0.03) and isoquercitrin (4.27 ± 0.02). In EELPa, isoquercitrin was not present, and it was the species that had the lowest quantification of phenolic compounds, as shown in Table 2. 

The amounts of caffeic acid and chlorogenic acid found in the species studied in this work corroborate the study by Chaowuttikul et al. [23], who investigated the determination of the two acids in 100 selected plants from several families, including Lamiaceae. The results of RP-HPLC analysis demonstrated that the distribution of these two phenolic compounds varied in many samples. Among 100 selected plants, 80.18 % contained all two compounds, 14.41% contained only one compound, and 5.41% could not detect these two compounds. Given the facts, we can suggest that chlorogenic acid (3.87 ± 0.02–4.51 ± 0.03) and caffeic acid (2.15 ± 0.01–9.76 ± 0.05) act as chemical markers in species of the genus *Plecthanthus*.

### 2.2. Antimicrobial Activity

Among all their ethnopharmacological applications, the species of the genus *Plectranthus* are reported to be used for the treatment of infections and as antiseptic agents, which is consistent with the literature reporting their potential as antimicrobials [24,25,26].

The extracts of the three *Plectranthus* species studied showed antibacterial capacity on both Gram-negative and Gram-positive strains, as shown in Table 3. The best antibacterial effects were evidenced for the *P. amboinicus* extract, with minimum inhibitory concentration (MIC) values reaching up to 16 µg/mL against Gram-positive strains, especially the multi-resistant *S. aureus*. This bacteria class is more susceptible to penetration because its cell wall does not have an extra membrane as the Gram-negative ones do [27].

Almeida et al. [28], analyzing the antimicrobial potential of medicinal species from the Caatinga and Atlantic Forest, obtained for the ethanol extract of *P. amboinicus* MIC of >2000 µg/mL against *S. aureus* and *B. subtilis*, showing a lower potential than that obtained in this study. The ethanol extract of the dried leaves of *P. amboinicus* and its ethyl acetate fraction were tested against 14 clinical isolates of *S. aureus* resistant to methicillin (MRSA), obtaining MIC ranging between 2000–4000 µg/mL and 500–250 µg/mL, respectively, also showing lower efficacy than that obtained in this study [29].

Mothana et al. [30], evaluating the antibacterial potential of *Plectranthus* species, demonstrated the inhibitory potential of *P. barbatus* and its hexanic, chloroformic, and butanolic fractions against Gram-positive strains with MIC ranging from 62.5 to 312.5 µg/mL, corroborating the results obtained. The aqueous and aqueous aceto extracts of *P. barbatus* leaves showed MIC against *E. coli*, *P. aeruginosa* and two strains of MRSA ≥ 500 µg/mL, showing lower effectiveness than the extract under study [9].

Silva and collaborators [31], studying the antimicrobial potential of plant extracts, obtained for ethanolic extract of *P. ornatus* leaves against two strains of *S. aureus* MIC of 1200 µg/mL. The dichloromethane extract from the leaves of this species, on the other hand, showed good antibacterial potential against *S. aureus*, with MIC ranging from 500 to 700 µg/mL [32]. These results show low potentials when compared to the results of this study.

The results of the direct contact modulatory activity assays show that the antibiotic activity against the aminoglycosides was potentialized in the presence of extracts. *P. amboinicus* potentiated the antibiotic activity of all drugs against all tested strains, as shown in Figure 2. The species *P. barbatus* potentiated gentamicin against *P. vulgaris*, *S. aureus* and multidrug-resistant *S. aureus*, decreasing all the MIC from 128 µg/mL to 16 µg/mL (Figure 3). The extract of *P. ornatus* produced a synergistic effect on kanamycin and gentamicin against *E. coli* and *S. aureus*, respectively, decreasing the MIC values from 128 µg/mL to 16 µg/mL in both cases (Figure 4).

Silva and collaborators [31], by studying plant extracts with antibacterial potential, demonstrated that the interaction between the ethanol extract of *P. ornatus* leaves and the antibiotics ampicillin, kanamycin, and gentamicin had a synergistic effect against two *S. aureus* strains, corroborating the results obtained in this study. On the other hand, the ethanol extract of *P. barbatus* leaves, combined with streptomycin, showed a synergistic effect against Gram-negative strains of *E. coli* and *P. aeruginosa* [33]. Although several studies report the antimicrobial activity of the genus *Plectranthus*, there is no research found employing crude extracts of *P. amboinicus* as modulator of the action of aminoglycoside class antibiotics.

The use of extracts and fractions as antimicrobial agents presents a low risk of increased resistance, because they are complex mixtures, making microbial adaptability difficult. Considering this variety in the chemical nature of their compositions, they present varied mechanisms of interfering with microbial growth by interacting with the lipid bilayer and facilitating the disruption of cellular activity [34].

Several combinations enable the conventional antibiotic to interact with its target inside the bacterial cell, and additionally, some compounds can act by other antimicrobial mechanisms. Synergistic interactions are a tool to expand the antimicrobial spectrum, prevent the emergence of new resistance, and minimize toxicity by using low concentrations of both agents [35].

### 2.3. Antioxidant Activity

The results show that all extracts showed DPPH free radical scavenging activity (Figure 5), revealing that EELPa was the least effective, with IC_50_ 117.4 μg/mL, followed by EELPb, with IC_50_ 37.20 μg/mL. In contrast, EELPo exhibited the highest inhibition potential with an IC_50_ 32.21 μg/mL, which may correlate with the higher amount of phenolic compounds presented in the extract. The positive control ascorbic acid showed higher antioxidant activity, with an IC_50_ 1.77 μg/mL. The results indicate that the three extracts tested presented higher IC_50_ than the ascorbic acid control, but at the concentration of 0.5 mg/mL, the inhibition potentials of the EELPb and EELPo extracts were higher than that of ascorbic acid, with percentages of 84.33, 88.66 and 83.85%, respectively.

Gomes et al. [36], studying the influence of annual seasonal variation on phenolic content, and antioxidant activity of *P. amboinicus*, obtained IC_50_ ranging from 85.04 to 220.37 μg/mL. Ethanol, methanol and chloroform extracts of *P. amboinicus* leaves from India showed different IC_50_, which were 59.5, 124.9 and 137.1 μg/mL, in the same order, on which the authors relate this potential to the content of phenolic compounds present in the extracts [37].

The ethanol extract of the aerial parts of *P. barbatus* and its chloroform fraction obtained DPPH inhibition percentages of 81.2 and 80.1% at the concentration of 1000 μg/mL, proving to be close to that obtained by ascorbic acid of 94.4% [30]. As for the ethanol extract, lyophilized decoct and acetate fraction of *P. ornatus* leaf decoct showed potential DPPH free radical scavenging, with IC_50_ of 67.04, 66.62 and 12.35 μg/mL, respectively [38].

Medrado et al. [39], studying the relationship between different culture types, chemical composition, and antioxidant activity of *P. ornatus*, demonstrated that cultures with higher caffeic acid content showed better DPPH inhibition responses, with IC_50_ between 26 and 39.7 μg/mL; ethyl acetate fractions from quercetin-rich *P. amboinicus* leaves, on the other hand, showed moderate DPPH sequestering activity, with IC_50_ ranging from 124 to 137 μg/mL [40].

The structure of phenolic compounds is directly related to their ability to scavenge free radicals and chelate metals by donating hydrogen or electrons, which is influenced by the number of hydroxyl groups and their positions in relation to the carboxyl group, chain saturation, glycosylation, and the presence of substituents on the rings. Furthermore, glycosylated phenolics exhibit weaker antioxidant activity than the corresponding aglycones; however, they are more bioavailable, which increases their antioxidant potency [41].

### 2.4. Chemometric Analysis

Principal component analysis, together with cluster analysis, was used to classify the species according to the content of the bioactive compound. The chemical PCA (Figure 6) showed that the first two PC accounted for the totality of variability, which was high enough to represent all the variables (58.5% for the PC1 and 41.5% for PC2). Using the bioactive compounds as chemical descriptors, it was clearly seen that the species were separated in the different quadrants in the PCA biplot. The inclusion of data from the antioxidant and antibacterial activity does not modify the separation; instead, it suggests that the antioxidant effect of these species presents a significant contribution to the synergistic effect of phytoconstituents, especially by the flavonoid contents such as rutin, catechin, quercitrin, kaempferol glycoside and phenolic acid such as gallic acid, chlorogenic acid, caffeic acid. The antimicrobial activity did not show significant variation among the studied *Plectranthus* species; however, the chemical of the phenolic acid (gallic acid, chlorogenic acid and caffeic acid) and flavonoid glycoside (rutin, quercitrin, kaempferol glycoside) presents a significant contribution to the antibacterial effect againt *S. aureus* ATCC 12692, isoquercitrin to *S. aureus* 358 and quercetin and kaempferol to *E. coli* ATCC 25922.

Our study was in agreement with other data in the literature, suggesting that antioxidant and antibacterial activity of *Plectranthus* extract was related to the presence of some these classes of compounds as gallic and caffeic acid [42]. The study by Jhanji et al. [43] showed that rutin had a greater inhibitory effect in *S. aureus* and the lowest effect against drug-resistant *E. coli* and *S. aureus*. Studies with phenolic acids present a positive correlation between increasing hydroxyl and methoxy groups with antioxidant properties (as observed in the positive value of PC2); however, there was a slightly decreased antimicrobial efficacy with increases number of compounds that present these groups [44]. A positive value of PC1 demonstrates the importance of these flavonoids and phenolic acid to antibacterial activity, which can act selectively on Gram-positive [45] and -negative bacterial cells [46], and the literature data show that these compounds can promote these compounds the inhibition of virulence factors as biofilm formation [47].

In this study, we examine the chemical composition of extracts from *Plectranthus* species (Lamiaceae) and establish a correlation between the antioxidant and antibacterial activities by utilizing chemometric methods. These data provide comparison basis and valuable information that contribute to the use of these extracts or isolate compounds as a possible therapeutic antibacterial agent.

## 3. Materials and Methods

### 3.1. Plant Materials

The leaves of *Plectranthus amboinicus* (Lour.) Spreng, *Plectranthus barbatus* (Andr.), and *Plectranthus ornatus* (Codd) were collected in the Horto de Plantas Medicinais e Aromáticas in Laboratório de Pesquisas de Produtos Naturais (LPPN) from Universidade Regional do Cariri (URCA), Crato, state of Ceará, Brazil. A voucher of each specimen was deposited in the Herbário Caririense Dárdano de Andrade—Lima (HCDAL), URCA, registered under No 3037, 3038, and 3039, for *P. amboinicus*, *P. barbatus*, and *P. ornatus*, respectively.

### 3.2. Preparation of Extracts

The extracts were prepared used fresh leaves (500 g) of the three *Plectranthus* species. The samples were added separately in flasks with 2 L of ethanol 95% (*w*/*v*) and left in ambient temperature (30 °C) for 72 h. The ethanol was removed using a rotary vacuum evaporator (Model Q-214M2, Quimis, Brazil) and ultrathermal bath (Model Q-214M2, Quimis) under reduced pressure at a temperature of 60 °C. The yields obtained for the crude ethanol extracts were 6.0%, 4.5% and 3.8% for *P. amboinicus* (EELPa), *P. barbatus* (EELPb) and *P. ornatus* (EELPo), respectively.

### 3.3. Chemical Characterization

#### 3.3.1. Phytochemical Screening

A solution containing 300 mg of each extract was diluted in 30 mL of ethanol 70% (*w*/*v*). Aliquots containing 3 mL of this solution were subjected to the addition of specific reagents such as: iron chloride, acetic acid, ammonium hydroxide, potassium dichromate, Dragendorff, among others. These qualitative tests are based on color or precipitation reactions as a positive response to the presence of specific classes of secondary metabolites, when in contact with reagents. All reactions allow only the presence or absence of the chemical classes to be determined, and not the amounts of them that are present in different extracts [48,49].

#### 3.3.2. Phenolics and Flavonoids Compounds by HPLC/DAD

##### Chemical, Apparatus, and General Procedures

All chemicals were of analytical grade. Methanol, acetic acid, gallic acid, caffeic acid and chlorogenic acid were purchased from Merck (Darmstadt, Germany). Catechin, quercetin, rutin, quercitrin, isoquercitrin, glicoside kaempferol and kaempferol were acquired from Sigma Chemical Co. (St. Louis, MO, USA). High performance liquid chromatography with diode arrangement detector (HPLC/DAD) was performed with a Shimadzu Prominence Auto Sampler (SIL-20A) HPLC system (Shimadzu, Kyoto, Japan), equipped with Shimadzu LC-20AT reciprocating pumps connected to a DGU 20A5 degasser with a CBM 20A integrator, SPD-M20A diode array detector and LC solution 1.22 SP1 software.

##### HPLC/DAD Analysis

Reverse phase chromatographic analyses were carried out under gradient conditions using C18 column (4.6 mm × 250 mm) packed with 5 μm diameter particles; the mobile phase was water containing 2% acetic acid (A) and methanol (B), and the composition gradient was: 5% (B) for 2 min; 25% (B) until 10 min; 40, 50, 60, 70 and 80% (B) every 10 min. This was following the method described by Sabir et al. (2012) with slight modifications. 

Extracts of *Plectranthus* were filtered through 0.45 μm membrane filter (Millipore) and then degassed by ultrasonic bath prior to use, the extracts of *Plectranthus* were analyzed at a concentration of 6 mg/mL. The flow rate was 0.8 mL/min and the injection volume was 40 μL. The sample and mobile phase were filtered through 0.45 μm membrane filter (Millipore) and then degassed by ultrasonic bath prior to use. Stock solutions of standards references were prepared in the HPLC mobile phase at a concentration range of 0.050–0.250 mg/mL catechin, quercetin, rutin and kaempferol, and 0.020–0.200 mg/mL for gallic, chlorogenic and caffeic acids. Quantification was carried out by integrating the peaks using the external standard method, at 257 nm for gallic acid, 280 nm for catechin, 325 nm for chlorogenic and caffeic acids, and 365 for quercetin and rutin. The chromatography peaks were confirmed by comparing its retention time with those of reference standards and by diode arrangement detector (DAD) spectra (200 to 600 nm). All chromatography operations were carried out at an ambient temperature (25 °C) and in triplicate.

Quantifications of the compounds were based on analytical curves of the reference standards. The limit of detection (LOD) and limit of quantification (LOQ) were calculated based on the standard deviation of the responses and the slope using three independent analytical curves, as defined by IHC [50]. LOD and LOQ were calculated as 3.3 and 10 σ/S, respectively, where σ is the standard deviation of the response and S is the slope of the calibration curve.

### 3.4. Antibacterial Activity

#### 3.4.1. Minimal Inhibitory Concentration Test (MIC)

The assay was performed with six bacterial strains from the Oswaldo Cruz Foundation (FIOCRUZ): *Escherichia coli* (ATCC 25922), *Proteus vulgaris* (ATCC 13135), *Bacillus cereus* (ATCC 33018), *Pseudomonas aeruginosa* (ATCC 15442), *Staphylococcus aureus* (ATCC 12692), and multidrug-resistant strain of *Staphylococcus aureus* (Sa 358). The minimum inhibitory concentration was determined by the microdilution method according to documents M-100 [51]. The extracts were initially solubilized in sterile distilled water and dimethyl sulfoxide (DMSO) at 1024 µg/mL. An amount of 100 µL of inoculum was distributed in each well of the 96-well microdilution plate; subsequently, serial dilutions of the extract solution were performed obtaining concentrations from 512 to 8 µg mL. The plates were transferred and incubated for 24 h at 35 ± 2 °C, an air atmosphere on the stove. The results were analyzed by colorimetric reaction after adding 25 µL of a resazurin solution (0.01%) to each well after incubation. The minimum inhibitory concentration (MIC) was defined as the lowest concentration of extract capable of inhibiting the growth of microorganisms. 

#### 3.4.2. Modulating the Action of Antibiotics

The modulating effect of the extracts was analyzed by combining them with the aminoglycosides amikacin, kanamycin, and gentamicin, according to the methodology proposed by Coutinho et al. [52]. The extracts were tested at subinhibitory concentrations (MIC/8). In each well of the microdilution plate, 100 µL of the solution containing BHI culture medium (10%) and 100 µL of the inoculum and solution of the extracts (1024 µg/mL) were distributed. Subsequently, serial dilutions of the antibiotics were performed obtaining dilutions concentrations from 1024 to 0.5 µg/mL. The plates were incubated at 35 ± 2 °C for 24 h an air atmosphere on the stove, and read by colorimetry by the addition of 25 µL of resazurin solution (0.01%).

### 3.5. DPPH Free Radical Scavenging

The free radical scavenging activity of the extracts was determined by the photocolorimetric 2,2-diphenyl-1-picrylhydrazyl (DPPH) method proposed by Choi, Lee and Kang [53], with modifications. Concentrations of 0.001; 0.0025; 0.005; 0.01; 0.025; 0.05; 0.1; 0.25; 0.35 and 0.5 mg/mL of the extracts were used. In 96-well ELISA plates, 20 μL of the extract concentrations, 80 μL of 95% ethanol and 100 μL of ethanolic solution of the DPPH radical (0.4 mM) were added. After 30 min of incubation at ambient temperature (30 °C) under the shelter of light, the reading was performed at 518 nm in a spectrophotometer (UV-Visible TR Reader Shimadzu). The positive control used was ascorbic acid, under the same conditions as the samples, and the blank in the absence of DPPH. The negative control consisted of 100 μL of DPPH (0.4 mM) and 100 μL of 95% ethanol. The percentages (%) of DPPH radical inhibition were calculated using the following formula, in which Abs means absorbance:% DPPH inhibition = {[(AbsControl − (AbsExtract − AbsWhite)] × 100%}/AbsControl

### 3.6. Statistical Analysis

In the statistical analyses, the results were expressed as mean (*n* = 3) ± standard error (S.E.P.M). The data obtained in the phenols and flavonoids quantification by HPLC/DAD, modulation of antibiotic action, and DPPH free radical scavenging were submitted to analysis of variance (ANOVA) followed by Tukey’s test. Results with *p* < 0.05 were considered significant. All analyses were performed using GraphPad Prism 8.0 software. The chemometric analysis by principal components analysis (PCA) was procedure using the Jamovi v.2.2.

## 4. Conclusions

The present study reports the analysis of the chemical constituents of the species extracts of the genus *Plectranthus*, revealing flavonoids, tannins, and saponins as the uncommon constituents found in the three species; as well as highlighting the predominance of chlorogenic acid in EELPa, with caffeic acid being the most abundant compound in EELPb and EELPo. Antibacterial activity on Gram-positive and Gram-negative strains was observed for all species, with EELPa performing best against Gram positive strains. In the modulation tests, the best results were also with EELPa, suggesting that the modulatory action was potentialized by the chemical constitution of this extract in comparison to the other extracts. In the antioxidant test, EELPo showed the highest capacity to scavenge the DPPH free radical. The PCA of the chemical composition of the species of *Plectranthus* separated in three different chemotypes. The *P. amboinicus* was correlated has important antioxidant, whereas *P. ornatus* and *P. barbatus* were correlated to having important antibacterial activities. Updates and new evidence on *Plectranthus* species obtained in this study may serve as the basis for further development of antimicrobial agents and antioxidant compounds.

## Figures and Tables

**Figure 1 molecules-26-07665-f001:**
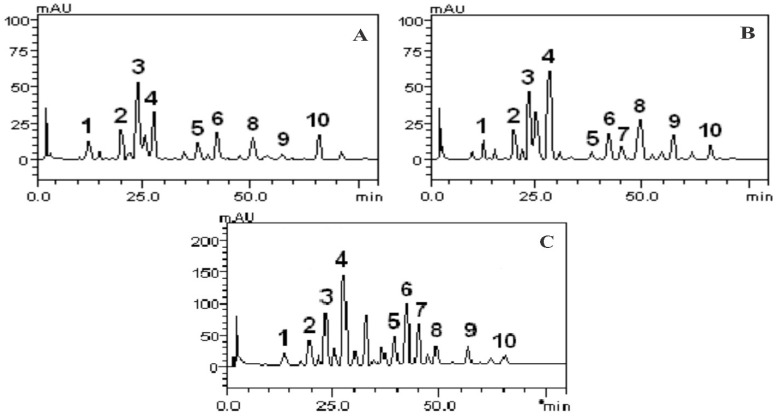
Elution profiles of HPLC/DAD of EELPa (**A**), EELPb (**B**) and EELPo (**C**). Gallic acid (peak 1), catechin (peak 2), chlorogenic acid (peak 3), caffeic acid (peak 4), rutin (peak 5), quercitrin (peak 6),isoquercitrin (peak 7), quercetin (peak 8), kaempferol (peak 9), kaempferol glycoside (peak 10). Calibration curve for gallic acid: Y = 16,479x + 1236.5 (r = 0.9991); catechin: Y = 11,355x + 1047.1 (r = 0.9987); chlorogenic acid: Y = 17,035x + 1304.6 (r = 0.9997); caffeic acid: Y = 13,674x + 1288.4 (r = 0.9989); rutin: Y = 14,756x + 1258.7 (r = 0.9999), quercetin: Y = 15,071x + 1241.6 (r = 0.9985), isoquercitrin: Y= 12,873x + 1325,6 (r = 0.9998); quercitrin: Y = 11,870x + 1329,8 (r = 0.9993) and kaempferol: Y = 12,953x + 1063.2 (r = 0.9997).

**Figure 2 molecules-26-07665-f002:**
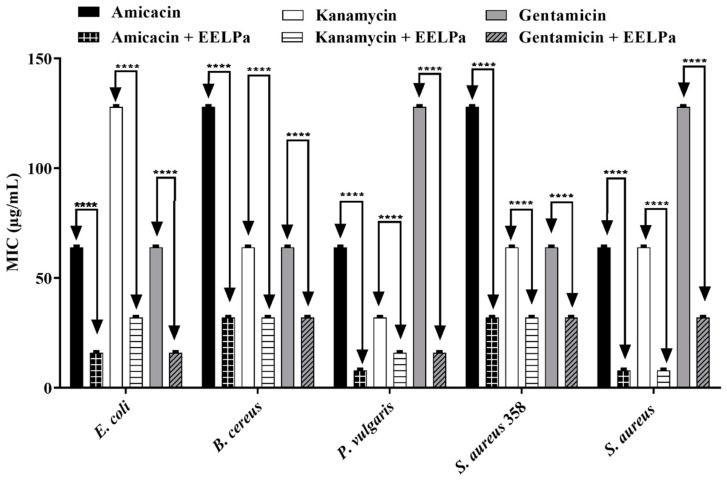
Modulating effect of ethanol extract of fresh *P. amboinicus* leaves on the antibiotic activity of amikacin, kanamycin and gentamicin against *E. coli*, *B. cereus*, *P. vulgaris*, *S. aureus* 358 and *S. aureus* strains. ****: *p* < 0.0001 (ANOVA and Tukey’s test).

**Figure 3 molecules-26-07665-f003:**
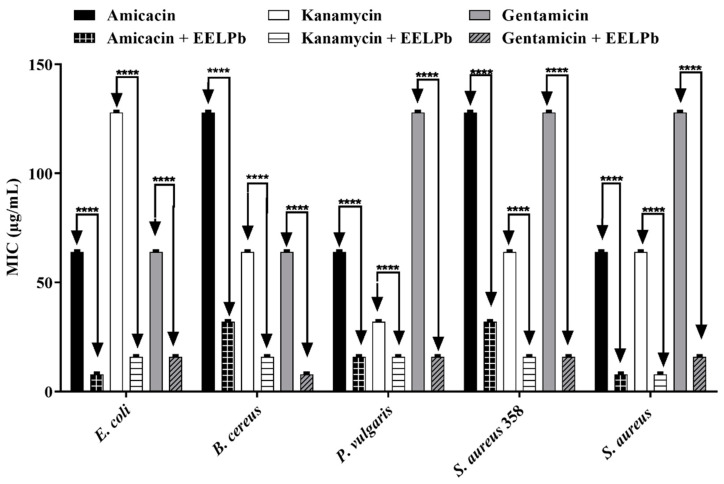
Modulatory effect of *P. barbatus* fresh leaf ethanol extract on the antibiotic activity of amikacin, kanamycin and gentamicin against *E. coli*, *B. cereus*, *P. vulgaris*, *S. aureus* 358 and *S. aureus* strains. ****: *p* < 0.0001 (ANOVA and Tukey’s test).

**Figure 4 molecules-26-07665-f004:**
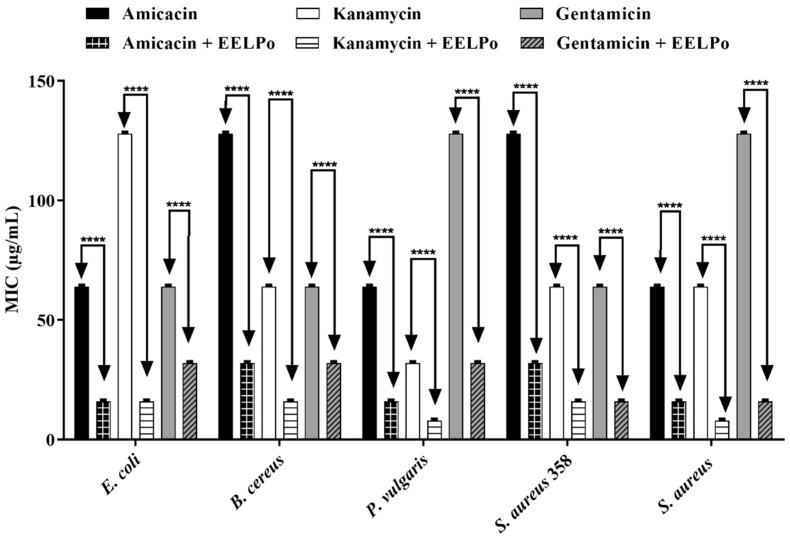
Modulating effect of *P. ornatus* fresh leaf ethanol extract on the antibiotic activity of amikacin, kanamycin and gentamicin against *E. coli*, *B. cereus*, *P. vulgaris*, *S. aureus* 358 and *S. aureus* strains. ****: *p* < 0.0001 (ANOVA and Tukey’s test).

**Figure 5 molecules-26-07665-f005:**
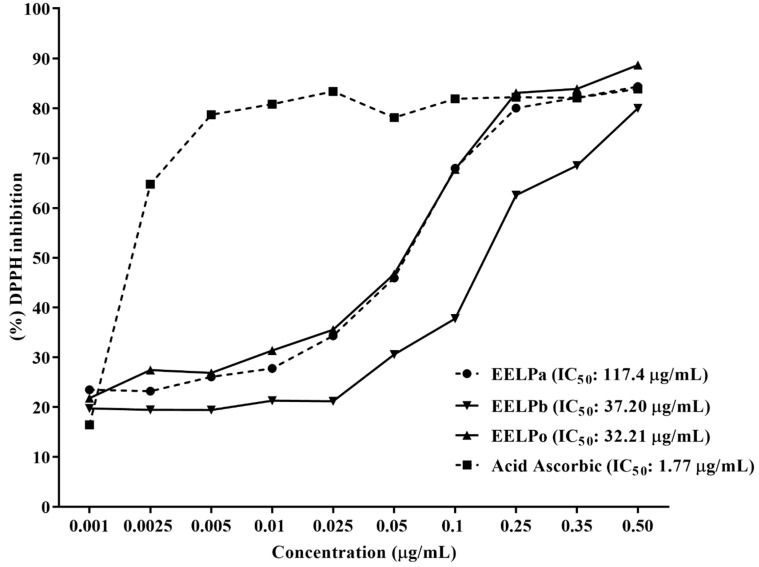
DPPH free radical scavenging by ethanol extracts of *P. amboinicus*, *P. barbatus* and *P. ornatus* fresh leaves and the positive control ascorbic acid. Nonlinear regression of the transformed curves (ANOVA and Tukey’s test).

**Figure 6 molecules-26-07665-f006:**
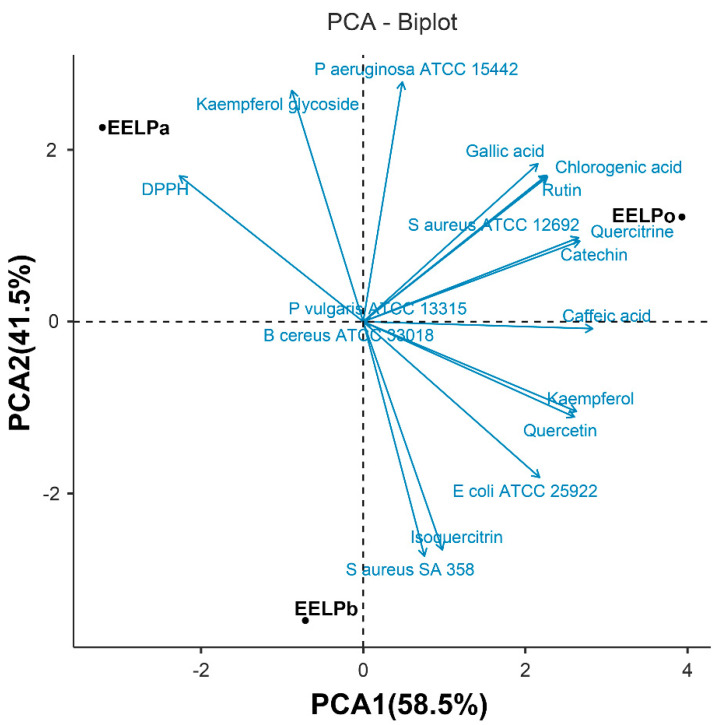
Principal component analysis (PCA) of the influence of flavonoids and phenolic acid phytoconstituents of *Plectranthus* species (Lamiaceae) under antibacterial and antioxidant activity.

**Table 1 molecules-26-07665-t001:** Secondary metabolites classes identified in the ethanol extracts of fresh leaves of *Plectranthus amboinicus*, *Plectranthus barbatus*, and *Plectranthus ornatus*.

Samples	Secondary Metabolites													
	CT	PT	TT	LA	AC	F	FV	FVN	XT	CH	AR	CQ	AL	SP
EELPa	+	+	+	-	+	+	-	+	+	-	+	-	-	+
EELPb	+	+	+	-	-	+	-	+	+	+	+	-	-	+
EELPo	+	+	+	-	-	+	-	+	+	+	+	-	-	+

CT: Condensed tannins; PT: Pyrogallic tannins; TT: Triterpenes; LA: Leucoanthocyanidins; AC: Anthocyanins and anthocyanidins; F: Flavones: FV: Flavonols; FVN: Flavonones; XT: Xanthones: CH: Chalcones; AR: Aurones: CQ: Catechins; AL: Alkaloids; SP: Saponins; EELPa: Ethanol Extract from *Plectranthus amboinicus* leaves; EELPb: Ethanol Extract from *Plectranthus barbatus* leaves; EELPo: Ethanol Extract from *Plectranthus ornatus* leaves. (+): Present; (-): Absent.

**Table 2 molecules-26-07665-t002:** Phenolic acids and flavonoids quantified by HPLC/DAD in ethanol extracts of *Plectranthus amboinicus*, *Plectranthus barbatus* and *Plectranthus ornatus* fresh leaves.

Compounds	EELPa		EELPb		EELPo	
	mg/g	%	mg/g	%	mg/g	%
Gallic Acid	1.13 ± 0.04 ^a^	0.11	0.74 ± 0.03 ^a^	0.07	1.85 ± 0.02 ^a^	0.18
Catechin	1.89 ± 0.02 ^b^	0.18	1.91 ± 0.01 ^b^	0.19	3.04 ± 0.01 ^b^	0.30
Chlorogenic Acid	4.06 ± 0.03 ^c^	0.40	3.87 ± 0.02 ^c^	0.38	4.51 ± 0.03 ^c^	0.45
Caffeic Acid	2.15 ± 0.01 ^d^	0.21	5.01 ± 0.01 ^d^	0.50	9.76 ± 0.05 ^d^	0.97
Rutin	1.09 ± 0.03 ^a^	0.10	0.26 ± 0.01 ^e^	0.02	3.12 ± 0.01 ^b^	0.31
Quercitrin	1.83 ± 0.02 ^b^	0.18	1.83 ± 0.03 b^f^	0.18	4.93 ± 0.03 ^e^	0.49
Isoquercitrin	-	-	0.91 ± 0.04 ^g^	0.09	4.27 ±0.02 ^c^	0.42
Quercetin	1.65 ± 0.01 ^e^	0.16	2.34 ± 0.01 ^h^	0.23	2.63 ± 0.01 ^f^	0.26
Kaempferol	0.28 ± 0.01 ^f^	0.02	1.76 ± 0.05 ^f^	0.17	2.45 ± 0.04 ^f^	0.24
Kaempferol glycoside	1.81 ± 0.02 ^b^	0.18	0.85 ± 0.02 ^g^	0.08	1.36 ± 0.03 ^g^	0.13
Total	15.89 ± 0.14 ^a^	1.37	19.48 ± 0.22 ^b^	1.91	37.92 ± 0.25 ^c^	3.75

Results of values are expressed as mean (mg/g of dry extract) ± S.P.E. (*n = 3*). Averages followed by different letters differ statistically (ANOVA, Tukey’s test at *p* < 0.001).

**Table 3 molecules-26-07665-t003:** Minimum inhibitory concentration (MIC) of ethanol extracts of *P. amboinicus*, *P. barbatus*, and *P. ornatus* fresh leaves.

Microorganism	MIC (µg/mL)		
	EELPa	EELPb	EELPo
*E. coli* ATCC 25922	64	128	128
*P. vulgaris* ATCC 13315	128	128	128
*B. cereus* ATCC 33018	256	256	256
*P. aeruginosa* ATCC 15442	256	128	256
*S. aureus* ATCC 12692	16	16	32
*S. aureus* SA 358	16	128	64

ATCC: American Type Culture Collection.

## Data Availability

Not applicable.

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
