# Peer review of "HPLC/DAD, Antibacterial and Antioxidant Activities of Plectranthus Species (Lamiaceae) Combined with the Chemometric Calculations"

_molecules, 2021, doi:10.3390/molecules26247665_

Round 1

Reviewer 1 Report

I suggest the following items/ points:

  • I recommend the following title: Phenolic compounds and Biological activities of Plectranthus species (Lamiaceae) combined with the chemometric calculations
  • Plz provide the full name for each abbreviations when mentioned at first time e.g DPPH in abstract part.
  • No data/information’s related to the obtained results in abstract section.
  • I ask authors for increase the available knowledge about the biological activities and phenolic compounds of Plectranthus species in Introduction section (the present part not enough as introduction to the manuscript title and objectives).
  • What is the concentration of ethanol solvent used for extraction of these species (plz mention in methods)
  • Sub-title no. 3.3 in methods need more details especially what is mean by acidic and basic 327 reagents.
  • Plz check the lines from 84-94, I think not strong related to the obtained results in sub-title 2-1
  • Plz transfer the lines from 102 to 109 into Introduction section.
  • I observed that retention time abbreviated in the current manuscript as rT, however I think it expressed as Rt.
  • Plz convert the scientific name for all plant species mentioned in this study to italic form
  • Plz revision the tested concentrations of antioxidant activity in methods (mentioned mg/L) but in Figure no. 5 mentioned as ug/L.
  • The authors mentioned at line no. 240 that “but at the concentration of 0.5 240 μg/mL” but this concentration not mentioned in methods part (the tested concentrations started from 1 ppm (ug/ml))

Reviewer 2 Report

I think that the manuscript entitled “HPLC/DAD, antibacterial and antioxidant activities of Plectranthus species (Lamiaceae) combined with the chemometric calculations" deserves publication in Molecules after major revision. The manuscript is very interesting, methodically, visually correctly executed. However, in my opinion, it requires thorough changes, especially the discussion of the results, which is practically non-existent. In addition, from about 150 lines, the Latin names of both plants and microorganisms were not spelled according to the Latin spelling rules. Recommends publishing the manuscript in the Molecules with necessary modifications.

Lines 3, 44: pleas change “Lamiaceae” into “Lamiaceae

Introduction: please complete the information and literature on bioactive compounds in Plectranthus

Line 65: please, write a discussion of the results obtained

Lines 72-101: This is not a discussion but an Introduction

Lines 81-83: Please check the names (spelling) of the ingredients

Lines 121-122: In my opinion this sentence should be in Methods ”Quantifications of the compounds by HPLC-DAD were based on analytical curves of the reference standards”

Pleas change in the manuscript “ethanolic extract” into “ethanol extract”

Lines 131 and 133: please check the information on the extracts “ethanolic extract” or “dry extract”

Table 2. what the “%” are represented? Whay is the sum not 100%?

Line 102: please, write a discussion of the results obtained “Quantification of phenolic compounds and flavonoids by HPLC-DAD”

Lines 103-115, 135-146, 154-165: This is not a discussion but an Introduction

Line 160: please change “in vitro and in vivo” into “in vitro and in vivo

Line 166: please, write a discussion of the results obtained “2.3. Antimicrobial activity”

Line 167-168; 170: please change “Plectranthus” into ” Plectranthus

Line 172: please change “P. amboinicus” into “P. amboinicus

Line 173: please change “S. aureus” into ”S. aureus

From line 179 to the end of the manuscript please write Latin names in italics

Figure 2 – 5: explain if it was used “ethanolic extract” or “dry extract”?

Lines 247, 259: pleas change :and colleagues” into “et al.”

Line 319: please complete the information on ethanol “%” “(v/v) or (v/w) or (…)”

Lines 319, 358, 395: please complete the ambient temperature information

Line 326: please complete the information on ethanol % “(v/v) or (v/w) or (…)”

Line 327-328: please complete the information on the “addition of acidic and basic reagents”. What exactly were the reagents, pH, concentration, course of the experiment, etc.

Line 328: please complete the information about the used equipment measure “color change and / or precipitate formation”

Line 334: please change “quercitrina, isoquercitrina” into “quercitrin, isoquercitrin”

Line 375 and 386-367: please complete information about atmosphere incubation

Line 406: please modify in the manuscript entry “p” and its value

There are many abbreviations in the manuscript, unfortunately not explained. At the beginning of the manuscript, I suggest using a list of abbreviations with an explanation.

Round 2

Reviewer 1 Report

No comments

Reviewer 2 Report

I think that the peer review manuscript entitled “HPLC/DAD, antibacterial and antioxidant activities of Plectranthus species (Lamiaceae) combined with the chemometric calculations" deserves publication in Molecules in present form.